# The Ecosystem Services and Green Infrastructure: A Systematic Review and the Gap of Economic Valuation

**Merve Ersoy Mirici** 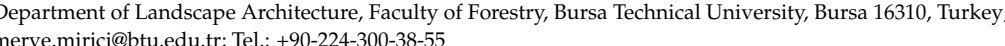

Department of Landscape Architecture, Faculty of Forestry, Bursa Technical University, Bursa 16310, Turkey; merve.mirici@btu.edu.tr; Tel.: +90-224-300-38-55

**Abstract:** This study was conducted to determine the trends at the intersection of studies made on green infrastructure and ecosystem services, which have frequently become preferred in establishing urban−green space relationships in global research. Green-related concepts have frequently been used from past to present in order to neutralise the increasing pressures on urban dynamics resulting from rapid urbanisation. Green corridor, green belt, green structure, and green finger/hand concepts have been used to provide recreational opportunities, protect nature, and keep urban sprawl under control. For the last decade, however, in addition to the traditional green concepts, green infrastructure (GI) and ecosystem services (ES) have been preferred in contemporary urban planning, as they enable the integration of the ecological concerns of the landscape and the socio-political perspective. The aim of this study is to detect the trends of the green infrastructure and ecosystem services association, and to reveal these trends in the common area with the bibliometric mapping method. The economic concept and its analysing use at the intersection of green infrastructure and ecosystem services were explored with VOSviewer using the Scopus® database. Furthermore, the number of documents, which initially began with around 39,719 studies, was reduced by filtering through systematic reviews, to only three documents that met the economic valuation criteria. In this way, a lack of economic analyses, creating a serious research gap within the framework of green infrastructure and ecosystem services, was quantitatively determined.

**Keywords:** green infrastructure; ecosystem services; economic valuation; VOSviewer; bibliometric analysis

## 1. Introduction

Economic growth on a global and local scale is an indispensable and irrevocable desire for every nation and society. Therefore, especially for emerging and developing countries, a dramatic increase in urbanisation is inevitable. It is predicted that approximately 70% of the world's population will live in cities by 2050 [1,2]. Rapid urbanisation is a major area of struggle for sustainable quality of life in cities [3]. Through economic actions and increasing population, urbanisation changes the local environment, and this change leads to environmental stress in urban residents. One of the biggest manifestations of this is the urban heat island effect. Long-term trends in many studies have shown that there is a relationship between urban centres and air temperatures, as the intensity of urbanisation increases through movement away from rural areas [4–6]. In fact, climate change may affect the climate dynamics of local regions [2]. Industrialisation, urbanisation, and transportation are the main driving forces of increasing greenhouse gas emissions caused by the anthropogenic effect [7]. While carbon neutral practices and reducing the impact of global climate change through the European green deal and the circular economy are on the agenda for industrialisation, it is imperative that measures are also taken for driving forces such as urbanisation and transportation. During the last two decades, green pursuits related to urbanisation, aimed at preserving the relationship between the biosphere and the city by coordination, and ensuring the stability and sustainable development of

urban landscapes, have come to the fore [8,9]. In fact, cities as a system involve nonlinear relationships and unpredictable complex behaviours [3]. It can be stated that in order to transfer unpredictable development to a sustainable basis, the green infrastructure and ecosystem services framework for urban landscape strategies often come to the fore. The common intersection of these two issues, which are independent of each other, is frequently the effort to create a foundation in urbanisation strategies, as, for the sake of urbanisation, virgin landscapes undergo transformation on different scales, resulting in a loss of ecosystem functions and biodiversity that affects human well-being [10].

During the last ten years, ecosystem services and green infrastructure approaches have begun to be dealt with more holistically in spatial planning practices involving urban systems. According to the European Commission, ecosystem services function as a complement to green infrastructure [11]. In urban areas, the components of green infrastructure are represented by a wide variety of ecosystem services and numerous habitat types [12]. In this way, it is possible to increase the adaptation and resilience of cities to climate change. Frequently, green infrastructure and ecosystem services are discussed within the scope of ecosystem-based adaptation. Adaptation to climate change is imperative in cities, where economic activities take place and where more than half of the world's population lives [2]. As part of the adaptation to global climate change, the ecosystem-based adaptation strategy in urban areas, beyond focusing merely on street trees and parks, offers the opportunity for more detailed research on understanding how human, ecosystem, and especially biodiversity sensitivity can be reduced through ecosystem services. In this study, the (i) density, (ii) trends, and (iii) research status of the economic on the intersection of green infrastructure and ecosystem services studies, which are becoming increasingly popular, are revealed by systematic review. In this review, the density of separate research into green infrastructure and ecosystem services, the density of mutual research, and the economic methods used in shareholder studies are revealed. The main motivation for this study is to determine that ecosystem services operate as an ecosystem function in the combination of green infrastructure and ecosystem services, and to reveal the deficiency in this direction in the conducted scientific studies. Bibliometric analyses are of great importance in terms of determining research trends, identifying scientific relationships in different research areas, and revealing the densities of these relationships [13].

## 2. Green Infrastructure and Ecosystem Services

Achieving the status of GI and ES research or planning studies as it is known today occurred through different process milestones in the last century. The GI idea is based on much earlier concepts like parkways, greenways, garden cities, green belts, or green wedges. The first planner initiator of greenways, which is one of the important components of GI, is known as Frederick Law Olmsted in United States [14,15]. Greenway movement came from emerald necklaces in the Boston green system, thereby the Greenway concept lies under the root of ecological connectivity of GI. The green belt was also associated with the garden city concept and, in the United Kingdom, with Ebenezer Howard. Garden city and green belts were developed as complementary to each other in the UK [16], but a green wedge was generated against rapid and extensive urban sprawl from Denmark. Stockholm's green wedges were first proposed by Sweden's Regional Planning Office in the 1990s (Figure 1). This radial structure allows for the formation of lengthy green wedges that serve several purposes: recreational, linking, and ecological [17–19].

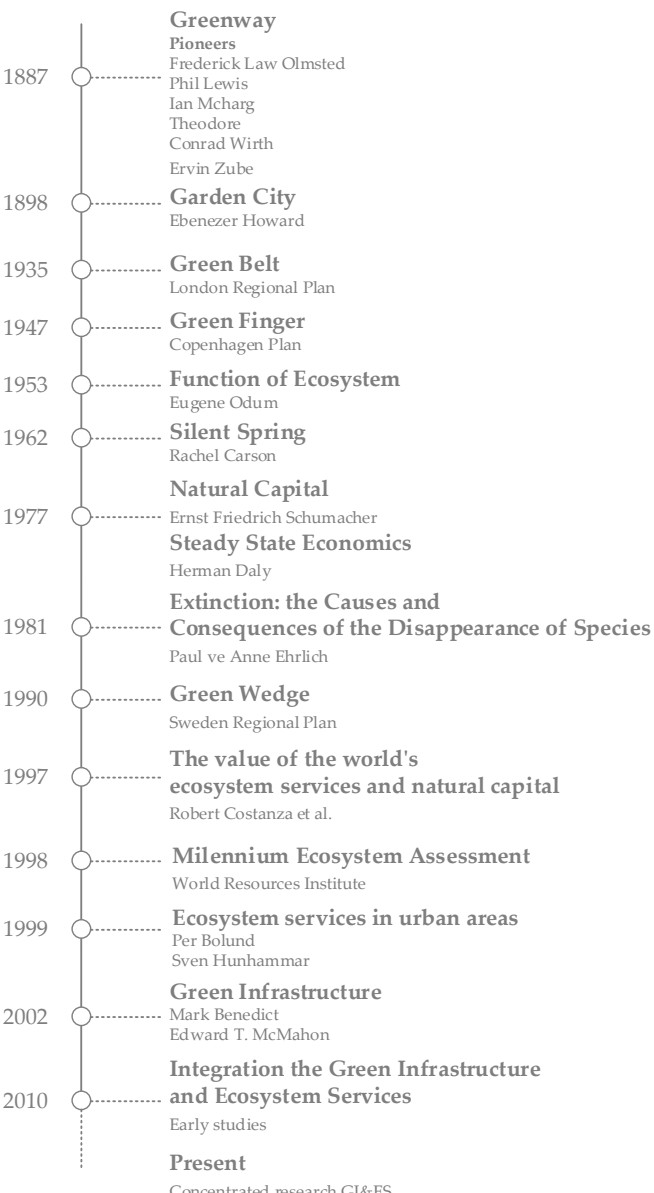

**Figure 1.** The milestones in green infrastructure and ecosystem services.

The green wedge has been undertaken as an emergency spare wheel against rapidly developing urban movement. In the meantime, the limit and concern of ecological resource/environmental crises has been presented. The ES framework emerged in order to economically put forward this ecological concern [20,21]. The emerging and developing GI and ES studies became integrated from 2010 on an urban scale [22].

*2.1. Green Infrastructure (GI)*

GI is regarded as a hybrid concept by some critics. The predecessors of this were the green belt drawn by landscape ecology, greenway planning and the garden city movement, water-based management plans, and the use of these components [23–25]. GI emphasises the focus on planning for biodiversity for conservationists [26], provision of social benefit for planners, watershed management and understanding of natural systems for engineers [27], and the socio-ecological benefit between the city/urban periphery for greenway experts [25]. Although handled in various ways among different professional disciplines, each interpretation and approach serve a non-spatial and universal theme. Especially at the end of the 1990s, a number of definitions were made about green infrastructure. However, the definition most often considered is as follows, "Green infrastructure, an interconnected

network of waterways, wetlands, woodlands, wildlife habitats, and other natural areas; greenways, parks, and other conservation lands; working farms, ranches, and forests; and wilderness and other open spaces that support native species, maintain natural ecological processes, sustain air and water resources, and contribute to the health and quality of life for people [28]". In addition, ref. [28] characterised the concept of GI as the harmony of coexistence between nature and humans. In essence, green infrastructure guarantees the maintenance of ecosystem services, promotes biodiversity conservation, enables the improvement and development of ecological connectivity, and implements measures to correct environmental imbalances in ecological restoration [29].

Instead of the passive position of nature in nature conservation, the concept of GI plays an active role by enabling coordination between man-made artificial constructions and nature protection [30]. This active role has increased the expectations towards the concept of green infrastructure, especially regarding the increasing effects of the urban heat island, the effects of global climate change, the increasing need for open green space after COVID-19, and transition to the new green order in Europe. Therefore, the GI concept has come to the fore, not only from the perspective of supporting biodiversity and protecting nature, but also that of people's processes of benefiting from the GI system. For this reason, the relationship between GI and ES has become more popular. Indeed, it has been revealed in many studies that the green infrastructure plays an important role in adaptation to climate change, improving storm water management capacity, reducing the urban heat island effect, reducing environmental pollution, improving socio-cultural facilities, serving social equality, increasing aesthetics, and improving social well-being [24,31–35]. In nature-based solutions, GI offers a wide spectrum of ecosystem services in the design and management of nature's strategically planned networks, semi-natural space, and other environmental components to ensure human health protection, welfare, and sustainable development [29]. It is a network system in which a wide range of opportunities, environmental features, and semi-natural and natural areas provided by ES are strategically planned under the umbrella of GI [36]. Therefore, green infrastructure and ecosystem services stand out as concepts that are often combined and that complement each other, as GI systems, especially in the urban environment, not only provide green roofs and street trees, but also provide ecological and economic benefits for city dwellers.

### *2.2. Ecosystem Services (ES)*

The increasing population and industrial developments in the last century have caused dramatic effects on ecosystem functions. The limited ecosystem resources that the world possesses and the increasing demands on the supply of these resources have revealed the need to conduct more detailed research into natural resources. Studies conducted in recent years have revealed the relationship between economic losses resulting from environmental disasters and the degradation of natural ecosystems [37,38]. In this direction, during the last ten years, ecosystem products and services as a landscape and urban regional planning tool have become an ever-growing field of research. While GI provides the bridge between nature and humans in the urban focus, the opportunities that the green infrastructure provides for human welfare are described as ecosystem services. GI areas provide various benefits by improving the quality of urban life via ES [39,40].

Directly and indirectly, the extent of people's benefit from the ecosystem is realised through ecosystem services [41,42]. Although ecosystem services are a research area generated in a single period and process, they have been shaped by many perspectives that have affected the ecosystem services framework from past to present. While the modern history of ecosystem services began in 1980 [20], the first estimate of ES on an economic basis on a global scale was made in 1997 [43]. Although a number of categories [44] have been established for ES, the Millennium Ecosystem Assessment has grouped ES categories under four main headings, as provisioning, regulating, supporting, and cultural services [45]. In the report prepared under the leadership of the Millennium Ecosystem Assessment, it was reported that the world ecosystem was degraded at a rate of approximately 60%. One of

the most important reasons for this degradation is anthropogenic effects. Loss of ecosystem services from land cover change alone has been estimated at US$ 4.3–20.2 trillion/year. Industrialisation, urbanisation, and transportation are the leading anthropogenic effects that damage ecosystem service potential [46]. Therefore, the destruction of ecosystem services first affects the national GDP and ultimately the global GDP [47]. The degradation and loss of ecosystem services sounds a serious alarm for human welfare, as well as regional and global eco-security [48]. In particular, rapid and unplanned urbanisation affects ecosystem service opportunities [49]. Therefore, green infrastructure issues, which enable bridge building between ecosystem services and urban dynamics, have begun to be studied with increasing momentum.

## 3. Materials and Methods

Although regarded as relatively new, the two fields of study, namely green infrastructure and ecosystem services studies, have grown considerably over the past two decades. Therefore, quantitative literature studies have been conducted separately in both fields of study: GI [30,50,51] and ES [51–53]. Although the concepts of these have begun to be used together in urban planning-based studies, very few bibliometric analyses of joint studies have been made. For this reason, in this study, the aim is, by emphasising this deficiency, to determine both the research trends of green infrastructure and ecosystem services studies on common ground and the status of production of these trends with economic outputs.

The approach in this study is based on the systematic determination of (i) quantitative information of publications related to the green infrastructure and ecosystem services separately, (ii) studies conducted under a common title, and (iii) to examine the economic valuation studies made on GI and ES association by using hierarchical filter (VOS mapping). The flow diagram in Figure 2 shows the different strategies and selection criteria. A large number of collected articles were first analysed by means of VOSviewer software for the qualitative article analysis. VOSviewer software is a freely available software tool that enables the generation of bibliometric maps based on networks of keywords. The bibliometric analysis in VOSviewer software implements the mapping approach, and based on co-occurrence data analysis, a similarity matrix is calculated [54]. The basic approach of the software is the visualisation of similarities (VOS). Eight filtering steps were applied in the study. Initially, quantitative findings and ultimately, qualitative findings were evaluated in the conducted studies. The publications selected as a result of the filtering were used for the qualitative evaluation.

*Filters Strategies*

In this study, the filter hierarchy consisting of the types of analysed documents—database, period of time, and keywords. Basically, three thresholds and eight filters are used in the filtering approach. The first threshold, keywords, was initially run separately for ES and GI. In this way, it was possible to compare qualitative densities with each other as different study areas. The second threshold was the determination of the densities and tendencies of the joint studies in these different fields of study. Finally, the third threshold was set as the determination of the extent of use of economic outputs, which is one of the basic requirements of the ES approach, in GI studies. In fact, in the last decade, the ecosystem services approach has been utilised by many researchers with an increasing intensity. However, through this study, the aim is to remark on the lack or gap of economic studies conducted within the framework of ES and GI, and the deficiency in this area. In fact, economic outputs are the most important of the main outputs of the ecosystem services area. Document types, namely peer review SCI or SCI-E articles, were evaluated by means of Scopus®. The filter hierarchy is given Figure 2 and Table 1.

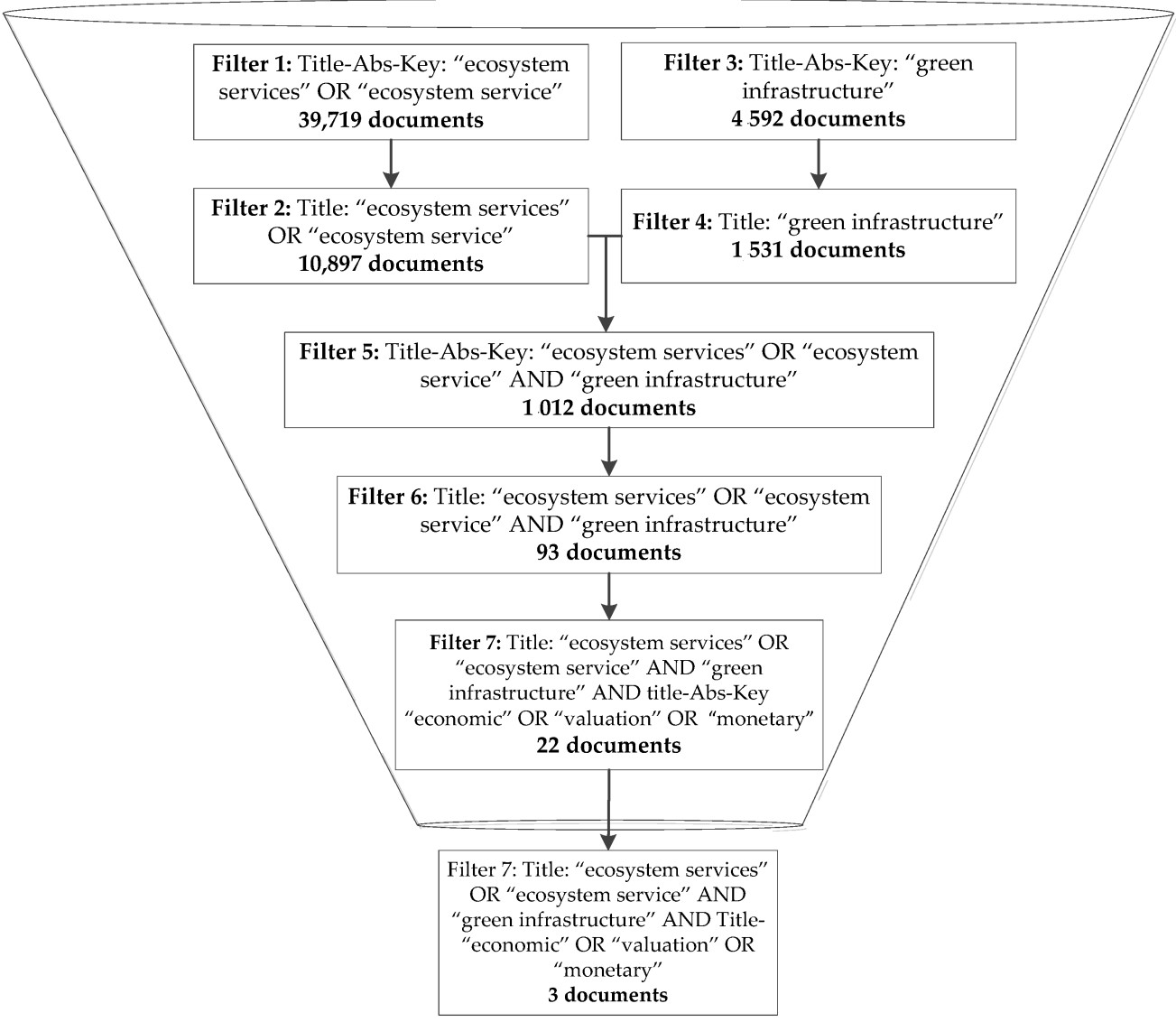

**Figure 2.** Workflow diagram in the study.

Scientific publications were downloaded in CSV format through filters in the Scopus®
database. The author name, year, document title, citation count, publisher, and index
keywords attributes of scientific publications were taken into account in the downloaded
CSV database. In the bibliometric analysis, a quantitative analysis approach and knowledge
mapping technique were used. Knowledge mapping through bibliometric analysis is
focused on connection strength and network, especially with keywords. Network analysis
generally facilitates the identification of trends and the readability of scientific studies by
means of clustering of keywords. The CSV database was uploaded to VOSviewer software
for bibliometric mapping. Relationship networks were determined by analysis of the
co-occurrence between keywords.

**Table 1.** Filter strategies.

| Field | Applied Filter | Field | Applied Filter |
|---|---|---|---|
| **Filter1** | | **Filter3** | |
| Keyword | "ecosystem services" OR "ecosystem service" | Keyword | "green infrastructure" |
| Search within | Title, abstract, keywords | Search within | Title, abstract, keywords |
| Time | to 2021 | Time | to 2021 |
| Database | Scopus | Database | Scopus |
| Result | 39,719 documents | Result | −4592 documents |
| **Filter 2** | | **Filter 4** | |
| Keyword | "ecosystem services" OR "ecosystem service" | Keyword | "green infrastructure" |
| Search within | Title | Search within | Title |
| Time | to 2021 | Time | to 2021 |
| Database | Scopus | Database | Scopus |
| Result | −10,897 documents | Result | 1531 documents |
| **Filter 5** | | | |
| Keyword | | | "ecosystem services" OR "ecosystem service" AND "green infrastructure" |
| Search within | | | Title, abstract, keywords |
| Time | | | to 2021 |
| Database | | | Scopus |
| Result | | | 1012 documents |
| **Filter 6** | | | |
| Keyword | | | "ecosystem services" OR "ecosystem service" AND "green infrastructure" |
| Search within | | | Title |
| Time | | | to 2021 |
| Database | | | Scopus |
| Result | | | 93 documents |
| **Filter 7** | | | |
| Keyword | | | Title: "ecosystem services" OR "ecosystem service" AND "green infrastructure" AND title-abs-key: "economic" OR "valuation" OR "monetary" |
| Search within | | | Title and title-abs-key |
| Time | | | to 2021 |
| Database | | | Scopus |
| Result | | | 22 documents |
| **Filter 8** | | | |
| Keyword | | | Title: "ecosystem services" OR "ecosystem service" AND "green infrastructure" AND Title: "economic" OR "valuation" OR "monetary" |
| Search within | | | Title |
| Time | | | to 2021 |
| Database | | | Scopus |
| Result | | | 3 documents |

## 4. Results

*4.1. Bibliometric Mapping with VOSviewer*

One of the points that this study draws attention to is that there has been a serious accumulation of ES studies since 1980. In Filter 1, 39,719 studies were detected, in which ES were mentioned not only in the title, but also in the keywords and the abstract. As determined in Filter 2, in approximately one third of these studies, 10,897 studies were found with ecosystem services mentioned in their titles only. Independently of this, in Filter 3, in which only green infrastructure was included in the title, keyword, and abstract, 4,592 studies were found, while in Filter 4, studies that focused on GI and had green structure mentioned in the title were determined in about one third of the studies. This indicates that although ecosystem services and green infrastructure concepts are not both directly focused on, these concepts are often solely referred to in an increasingly popular way in environmental studies. In addition, this study focuses especially on the examination of studies in which ES and GI approaches are used together and directly focused on, and also on the status of study of the economic aspect, which is one of the main outputs of ES, in these studies. In Filter 6, the bibliometric mapping technique was used via VOSviewer in studies where both the concepts of "ecosystem services" and "green infrastructure" were used only in the title content. By using the bibliographical data index keywords in these 93 documents, it was possible to develop a network through co-occurrence links. The keywords and their links are mapped in Figure 3.

In the co-occurrence type analysis, each different colour presented in the network map in Figure 3 shows clusters representing strong relationships. Red clustered urban planning has the highest number of occurrence with nine times; followed by spatial planning and urban green infrastructure, which are both green clustered and with six times occurrence; then blue clustered multi-functionality with four times occurrence; and orange clustered landscape connectivity with three occurrences. The network nodes' colours were assigned by default by the software. In the density map colour scheme, dark yellow corresponds to highest item density and blue corresponds to the lowest item density. In the database obtained from the Scopus database, 10 different clusters containing at least two recurrences were obtained. In the cluster analysis of the 93 scientific documents focusing on (i) ecosystem services and (ii) green infrastructure, it was observed that (iii) cities, (iv) urban forestry, (v) urban and cultural services, (vi) sustainability and biodiversity, (vii) spatial planning, (viii) multi-functionality (landscape/city/climate), (ix) connectivity/corridor, and (x) land dynamics came to the fore.

According to Filter 6, the first study in which ecosystem services and green infrastructure were studied together began in 2010 as a single study [22]. In addition, the temporal network relationship of the networks and its density map are shown in Figure 4.

In Figure 4, colours are defined by the average publication per year of each keyword, with yellow representing the most recent and dark blue representing the oldest. It is possible to observe the average times of scientific publications and the current studies between 2017 and 2020 in Figure 4. In the average year of publication of each keyword index, yellow indicates the most current and recent research, while studies in dark blue are the oldest focuses of research. Accordingly, it was determined that, especially after 2019, academic studies were oriented towards current research topics such as urban green infrastructure, regulation, cultural ecosystem services, co-benefits, and landscape connectivity.

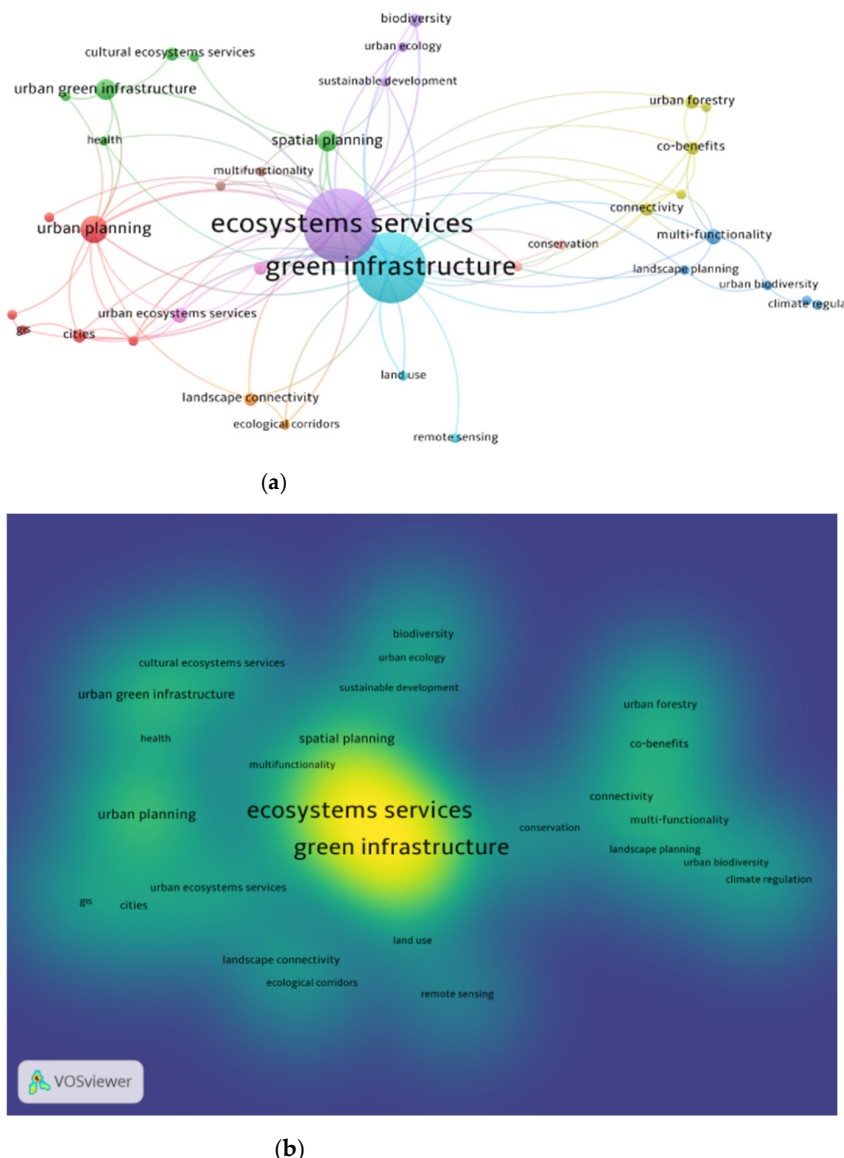

(**a**)

(**b**)

**Figure 3.** (**a**) Network map (nodes) and (**b**) density map of index keywords between ecosystem services and green infrastructure.

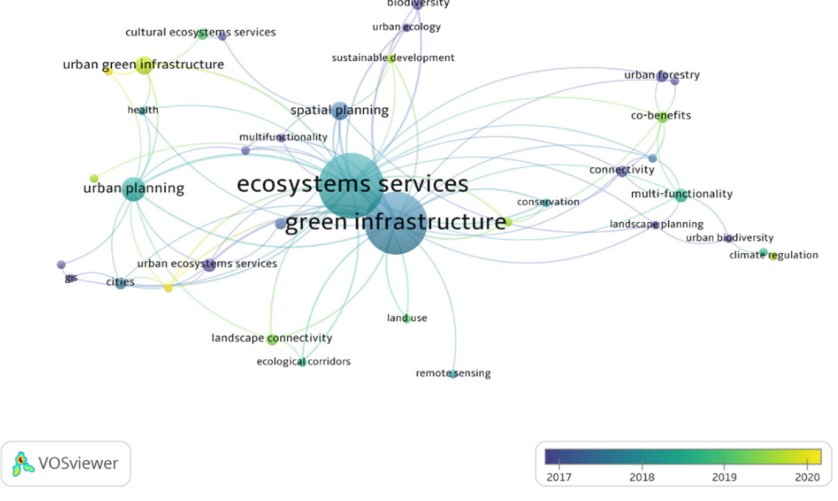

**Figure 4.** Time periods of the index keywords.

*4.2. Qualitative Review*

According to the elimination, applied as Filter 7, for studies in which GI (title), ES (title), and economic valuation (title-abs-keywords) concepts were studied jointly (Table 1), only 22 documents were identified from the past to the present. The first research in the intersection of this trio was made in 2010 [22].

Figure 5 shows the distribution of these studies between the 2010–2021 period. A slight gradual increase in joint studies was detected from 2010 to 2021 (Figure 5). In this study, Filter 7 is the intersection of GI (*title*), ES (*title*), and economic valuation (*title-abs-keywords*). The study numbers in this trio research area are inconsistent, while articles on GI and ES have been continuously increasing [30,55]. This filter found 22 studies, which are shown in Appendix A. The number of publications seems at an appropriate level, but the economic outputs of these publications are quite low. According to Filter 7 and Appendix A, only 6 of 22 publications performed an economical valuation analysis. In the abstract or keywords, "economic" is used in general terms and just to emphasize, and thus causes a fairly misleading situation. A great number of publications include "economic" frequently, using the general patterns such as economic benefit, economic growth, economic planning, or economic profile [11,56–71]. For this reason, we added Filter 8 to see more transparently the economic valuation outputs of GI and ES together. Finally, Filter 8 found three publications, but just [22] and [57] are peer-reviewed journal articles, and [56] is a conference paper. These three publications—(i) Addressing the information gaps associated with valuing green infrastructure in west Michigan: INtegrated Valuation of Ecosystem Services Tool (INVEST) (2010) [22]—drew attention to the increase in grey infrastructure systems and the reduction in green infrastructure systems caused by rapid urbanisation and population growth in Western Michigan. It was argued that due to the reduction in green infrastructure caused by this urbanisation and population movement, the benefits that people derive from natural ecosystems decrease. Stating that it is not possible for ecosystem services related to cities to be evaluated with traditional commercial markets, the study used the INtegrated Valuation of Ecosystem Services Tool (InVEST). The study strongly emphasised the economic focus, claiming that InVEST was first developed with the aim of educating local decision makers on the economic value of green infrastructure and ecosystem services in West Michigan. It was particularly emphasised that policy-makers were either unaware of green infrastructure and economic interconnectedness, or that this information was not available. Some of the 11 ecosystem services selected in the study were determined as market-based, while others were determined as non-market-based services. Benefit transfer was used in the economic valuation method. Accordingly, it was estimated that Western Michigan provided an annual green infrastructure value of 1.7 billion dollars. (ii) Another study resulting from Filter 8 was a congress presentation by [56]. The focus was on the economic contribution of green roof systems to the urban ecosystem in Melbourne, Australia's second largest city. It was stated that ecosystem services especially reduced the urban heat island effect, improved air quality, enabled energy saving, increased adaptation to climate change, increased habitat, and made a positive contribution to community liveability. In the hypothetical case study, it was estimated that 300 square metres of green roof reduced the runoff of approximately 93 kilolitres of rainwater per year, resulting in an annual economic benefit of 1245 AUD. The calculation of the economic benefit was enabled by the adaptation of formulae belonging to previous studies. (iii) The final document obtained in Filter 7 was made for New York by [57]. However, in this study, although ecosystem services valuation is mentioned in the keyword indexes, an economic analysis was not made. In this study, the five different ecosystem services were analysed with the multicriteria method. No economic output was generated. Prominently, it can be appropriate to show just one publication according to Filter 8 [22].

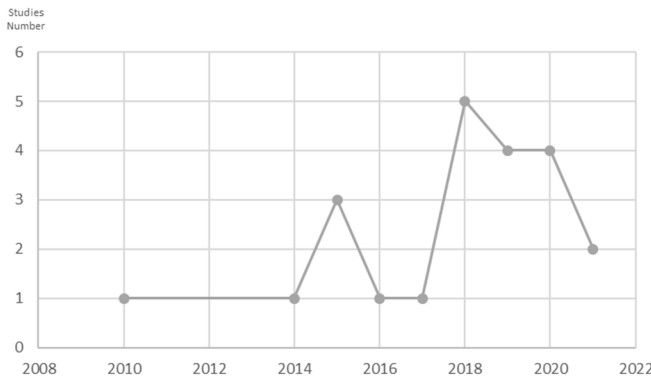

**Figure 5.** The histogram representing Filter 7 (22 studies).

## 5. Discussion

It has been observed that the studies in which GI and ES frameworks began to intersect date back to the last two decades. In the investigation made in Filter 5, approximately 1012 studies from the past to the present were identified. It is possible to observe in Figure 6 that the density of these studies increased significantly, especially after 2015.

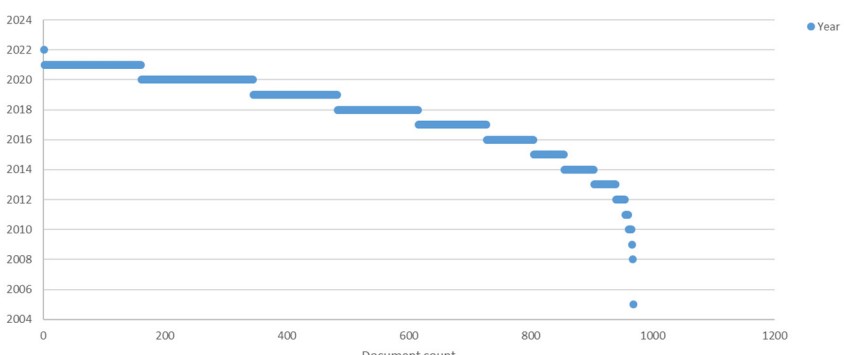

**Figure 6.** The histogram of common studies in GI and ES.

When examining the publications in which GI and ES are discussed in the title, keywords, and abstracts in the same study, it is possible to observe that there is an increasing trend for a close association in both frameworks. One of the main reasons for this could be the increasing dominance of urban dynamics. In fact, although GI is a framework built especially within urban systems, the absence of urbanisation in the desired green deal has triggered a new search, as rather than dealing with the city with a purely green conception, the utilisation of ecosystem services that enrich the content of GI is enabled. ES have a very wide range. Therefore, the union of GI and ES has become imperative for GI in order to be seated on a more solid foundation. The main question that this study attempts to draw attention to focuses on whether or not ecosystem services create an economic output. Looking at the history of ES, it has been emphasised that the ecosystem is not a free good, and that it must necessarily be given an economic valuation [39]. Measurability in economic, health, and spatial planning policies and strategies is required for protecting urban systems and biodiversity. The fact that energy and natural resources for producing products and services not only threaten natural habitats, but that this threat also has an impact on human well-being, is a great indicator [50]. Traditional economics has not evaluated these environmental services. Natural ecosystems have a protection cost. Furthermore, it has been shown in recent studies that this economic cost has an effect on human health and welfare of between 10 and 100 times [72–74].

In fact, by drawing attention to it in this study, the point made is that although GI and the ES framework have been studied very frequently and intensively, the fact that economic findings, which are one of the basic requirements of ES, are not studied, is

revealed as a major problem. In about 1,012 studies in which GI and ES began to be mentioned together, only two studies included economic findings. Considering that one of these studies was a conference presentation, it can be stated that economic outputs are included in the association of GI and ES, with only one article.

## 6. Limitation of Study

We compiled publications from 1984 through 2021 using the Scopus® database. On 18 December 2021, we downloaded the data and searched all articles containing SCI and SCI-E. Web of Science (WoS) and Scopus are the two main bibliographic databases [75]. For more than 40 years, WoS was the only source of bibliographic data until 2004, when Elsevier published Scopus [76]. Despite the fact that the WoS and Scopus databases have been widely compared for over 15 years, the scientometric community has not reached a conclusion yet on "which one is superior". However, the key distinction from WoS is that Scopus provides access to all of its information via a single subscription and Scopus also includes content from many specialized databases [75–78]. The WoS database have not the analysis before 2009, such as Urban Forestry and Urban Greening [62]. For this reason, in this study, we used the Scopus database for our investigation because it is more user-friendly and is better suited for reviewing study outcomes. The study is limited to SCI and SCI-E articles by means of Scopus. A few studies, besides the ones mentioned in this study, that focused on combining green infrastructure, ecosystem services, and economic valuation or non-SCI/SCI-E reports can be found via various platforms.

## 7. Conclusions

Many studies in the last decade have frequently started to mention GI and ES together in urban and smart city projects. This situation generally stands out as a strong argument in green space planning of local governments. In this study, the coexistence trends of the GI/ES research subjects were determined, and also, the lack of economic research, which creates a serious gap in this field, was quantitatively revealed. In addition, by using a total of eight filters by means of a systematic literature review, only three studies were identified on the GI/ES/economy axis in the Scopus® database. An economic valuation analysis was performed on only two of these three studies. While the main focus in the intersection of GI/ES indicates the urban planning area, the ecosystem services provided by landscape connectivity are frequently referred to in current studies.

(1) In this study, it has been shown that a serious accumulation of research has occurred, especially in the field of ES. Today and in the future, the differentiation of ES under specific categories can be enabled.

(2) While it is observed that cultural services come to the fore at the intersection of GI/ES, the expansion of regulation, support, and supply services can also be enabled.

(3) GI provides a criterion for minimising or avoiding the impact of urbanisation. The effort of this criterion to keep the ecosystem in balance by providing homeostasis on different axes constitutes the greatest determinant of ecosystem services. The common language of a wide range of different services is economic value, which is a common unit. It has been determined that the economic valuation processes, which are one of the main starting points of ES, have been largely ignored and there is a big gap in the research on this subject. In this direction, it has been revealed that urban GI systems are also an area that can be studied for economic analyses. It can be possible to develop this field by ensuring a qualitative increase in economic studies, which is one of the most important components of ecosystem services.

(4) The GI studies of today were reached by evolving from greenway, garden city, green belt, green finger, and green wedge concepts. Undoubtedly, these will still be evolving with ES and economic valuation into the future. In this respect, it is important that GI and ES are considered seriously in popular research.

(5) This study showed that publications should be examined not only quantitatively but also qualitatively, because we found 22 publications in Filter 7, but just 6 of them

generated economical outputs. Similarly, in Filter 8 (GI/ES/economic valuation), it three publications were found, but one of them was a conference paper and in the other publication no economic output was generated. The studies including GI/ES/economic valuation (under same title), one publication made in 2010 [50], is the first and only publication that fulfils this criteria in the Scopus ® database. The lack of research in this area can be addressed.

(6) Otherwise, economic value, economic valuation, valuation monetary, and trade-off terms make it difficult to find desired publications in ecosystem services. Instead of these terms, in order to prevent complexity, a unique word can be developed.

**Funding:** This research received no external funding.

**Data Availability Statement:** Data has been obtained from Scopus®.

**Acknowledgments:** The author would like to thank Ugur Avdan and Scopus ® database.

**Conflicts of Interest:** The author declare no conflict of interest.

## Appendix A

**Table A1.** Publication under the Filter 7 by Scopus ®.

| No | Ref. | Economic Valuation Method | Unit | Journal | Country | Year |
|----|------|---------------------------|------|---------|---------|------|
| 1 | [79] | Replacement cost<br>Carbon tax<br>Shadow project<br>Afforestation cost<br>Market price | Yuan | Sustainability | China | 2021 |
| 2 | [3] | The table equivalent value per unit area of ecosystem in China on Costanza's method | Yuan | Journal of Cleaner Production | China | 2021 |
| 3 | [80] | The willingness to pay | Dollar | Urban Forestry and Urban Greening | China | 2020 |
| 4 | [58] | Economic Valuation Not Performed | X | Sustainability | Italy | 2020 |
| 5 | [59] | No economic value | X | Ecosystem services | The Netherlands | 2020 |
| 6 | [60] | No economic value | X | Land | Italy | 2020 |
| 7 | [61] | No economic value | X | Ecosystem services | USA | 2019 |
| 8 | [62] | No economic value | X | Science of Total Environment | Spain | 2019 |
| 9 | [63] | No economic value | X | Urban Forestry and Urban Greening | Colombia | 2019 |
| 10 | [81] | Avoided cost for damages | EUR | Urban Forestry and Urban Greening | Italy | 2019 |
| 11 | [64] | No economic value | X | Landscape and Urban Planning | UK | 2018 |
| 12 | [65] | No economic value | X | Landscape and Urban Planning | South Africa | 2018 |
| 13 | [82] | Market price | EUR | Energy policy | Italy | 2018 |
| 14 | [11] | No economic value | X | Ekologia Bratislava | Slovakia | 2018 |
| 15 | [66] | No economic value | X | Belgeo | Russia | 2018 |
| 16 | [67] | No economic value | X | Sustainability | Poland | 2017 |
| 17 | [68] | No economic value | X | Land Use Policy | Finland | 2016 |
| 18 | [57] | No economic value | X | Environmental Science and Policy | USA | 2016 |
| 19 | [69] | No economic value | X | Environmental Science and Policy | Italy | 2015 |
| 20 | [70] | No economic value | X | Journal of Urban Planning and Development | Germany | 2015 |
| 21 | [71] | No economic value | X | Landscape Ecology | Italy | 2015 |
| 22 | [83] | Avoided cost<br>Replacement cost<br>The willingness to pay<br>Hedonic pricing<br>Contingent valuation | EUR | Building and Environmental | The Netherlands | 2014 |
| 23 | [22] | Benefit transfer | Dollar | Journal of Great Lakes Research | USA | 2010 |

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
