# Peer review of "The Ecosystem Services and Green Infrastructure: A Systematic Review and the Gap of Economic Valuation"

_sustainability, doi:10.3390/su14010517_

Round 1

Reviewer 1 Report

The paper presents an interesting approach to bridge studies in trending topics and filtered by the economic aspect in green infrastructure and the ecosystem services assessment. However, the reference to only one database limits the approach and the choice of that specific database is not clear and justified. The author's choice could be very briefly explained. 

The network map on Fig. 2 is nice, however the colour scheme (what does  blue color mean, e.g.) isn't clear and maybe a legend could improve the colors used. Does the distance between the keywords mean something? If yes, then a brief explanation could be given.

In addition, the outcomes could be improved additionally by strong recommendations and to conclude with more focused discussion on how the specific research field can be improved, if improvement is needed at all, etc.

The study highlights a nische outlook and represents the interlinks between green infrastructure and eccosystem services in an interesting way. I recommend the author to consider the abovementioned comments that can improve the quality of the work.

Author Response

Dear review, please see the attachment. 

Additional revised

  • All filters are revised as a recent date. This reason the document number increased. Fig 2 and Table 1 changed. But fig 3 did not change because according to filter 6 document number just 4 studies increased. That did not change trends and network maps.
  • Title 5 has been added as a limitation of studies.
  • The conclusion has been expanded.
  • GI and ES have been used Instate of green infrastructure and ecosystem service. The language union has been provided.
  • The reference list has been edited and added 34 new references

Reviewer 2 Report

Greetings,
Since it is a review paper, it is necessary to increase the number of references in the paper. Also, it is necessary to make an overview of these papers in paper. It is necessary to make tables for both Green infrastructure and Ecosystem services where you will list previous research, who did it, what it did and in which country the research was done. The Conclusion should also provide guidelines for future research.
All best.

Author Response

Dear review, please see the attachment. 

In addition to your suggestion;

  • All filters are revised as a recent date. This reason the document number increased. Fig 2 and Table 1 changed. But fig 3 did not change because according to filter 6 document number just 4 studies increased. That did not change trends and network maps.
  • Title 5 has been added as a limitation of studies.
  • The conclusion has been expanded.
  • GI and ES have been used Instate of green infrastructure and ecosystem service. The language union has been provided.
  • The reference list has been edited and added 34 new references

Reviewer 3 Report

Paper: The Ecosystem Services and Green Infrastructure: A Systematic Review and the Gap of Economic Valuation

Overview and general impressions

The topic is significant and relevant as green infrastructure is on the EU and the global urban agenda. The content of the paper informative and well supported with literature review on green infrastructure and ecosystem services. However, a great potential to improve the quality of the paper exists.

Discussion

Comments and questions:

  • The author states “… through this study, the aim is to reveal the quantity of economic studies conducted within the framework of ecosystem services and green infrastructure and the deficiency in this area” (p.4 lines 192-194) This sounds as a main aim which corresponds to the title of the paper. The structure and the content however, focuses on the content analysis without implementing the filter explicitly focused on “economic valuation “. Maybe this will significantly change the results.
  • Maybe it would be more valuable to outline different periods (as a result of the bibliometric study) and the trends and the interest to specific topics related to the GI and ES (as it changes with the changing frameworks and evidence-based policy development through the recent 30 decades).
  • There is also a need to justify the use of the Scopus® database. Is this a limitation of the research or a strive to high level peer-reviewed excellence papers? What about other databases? (See the attached file with suggested literature – evidence for research and publications on economic valuations of GI and ES. This literature offers relevant examples from databases outside Scopus and they generally do not support the hypothesis of the paper. What would be the recommendation about linking research to practise, reflection/evaluation of the implemented policies?
  • Are there publications on similar previous or recent qualitative studies (the attached file contains a link to a similar paper that uses the WoS database)?

Small warnings:

  • Page 2, line 68 _ please reconsider the word order, as it is not clear that all the mentioned 3 parameters density, trends and research status refer to the economic aspect of the GI.
  • Page 8 line 270 – there’s something wrong with the beginning of the sentence

Conclusion:

I came away with comments that restrict me from recommending this paper for publication as it stands. Therefore, I recommend that a revision should be warranted. I would ask that the author specifically address each of my comments.

Author Response

(The authors gave the same response as above.)

Round 2

Reviewer 2 Report

Greetings,
The authors corrected the paper in accordance with the reviews. The paper should be accepted now.
All best

Reviewer 3 Report

Most of my previous recommendations have been taken into account and significant effort has been made in order to improve the relevance and the quality of the research methodology.